# Age-Dependent Changes in the Effects of Androgens on Female Metabolic and Body Weight Regulation Systems in Humans and Laboratory Animals

**DOI:** 10.3390/ijms242316567

**Published:** 2023-11-21

**Authors:** Takeshi Iwasa, Hiroki Noguchi, Risa Tanano, Erika Yamanaka, Asuka Takeda, Kou Tamura, Hidenori Aoki, Tatsuro Sugimoto, Hikari Sasada, Takaaki Maeda, Saki Minato, Shota Yamamoto, Hiroaki Inui, Tomohiro Kagawa, Atsuko Yoshida, Ayuka Mineda, Mari Nii, Riyo Kinouchi, Kanako Yoshida, Yuri Yamamoto, Takashi Kaji

**Affiliations:** 1Department of Obstetrics and Gynecology, Institute of Biomedical Sciences, Tokushima University Graduate School, 3-18-15 Kuramoto-Cho, Tokushima 770-8503, Japan; noguchi.hiroki@tokushima-u.ac.jp (H.N.); tanano.risa@tokushima-u.ac.jp (R.T.); yamanaka.erika@tokushima-u.ac.jp (E.Y.); takeda.asuka@tokushima-u.ac.jp (A.T.); toku50302track@gmail.com (K.T.); aoki00q@gmail.com (H.A.); tsugimoto0608@gmail.com (T.S.); bjrsa5813@gmail.com (H.S.); sikakajyositakaaki@yahoo.co.jp (T.M.); minato.saki@tokushima-u.ac.jp (S.M.); c202290001@tokushima-u.ac.jp (S.Y.); hiroaki.inui@tokushima-u.ac.jp (H.I.); kagawa.tomohiro@tokushima-u.ac.jp (T.K.); yoshida.atsuko@tokushima-u.ac.jp (A.Y.); mineda.ayuka@tokushima-u.ac.jp (A.M.); nimari722@gmail.com (M.N.); kinouchi.riyo@tokushima-u.ac.jp (R.K.); yoshida.kanako@tokushima-u.ac.jp (K.Y.); yamamoto.yuri@tokushima-u.ac.jp (Y.Y.); kaji.takashi@tokushima-u.ac.jp (T.K.); 2Department of Renal and Genitourinary Surgery, Graduate School of Medicine, Hokkaido University, Sapporo 060-0808, Japan

**Keywords:** androgen, estrogen, PCOS, hypothalamus

## Abstract

In recent years, the effects of androgens on metabolic and body weight regulation systems and their underlying mechanisms have been gradually revealed in females. In women and experimental animals of reproductive age, androgen excess can adversely affect metabolic functioning, appetite, and body weight regulation. In addition, excess androgens can increase the risk of metabolic disorders, such as obesity, insulin resistance, and diabetes. These unfavorable effects of androgens are induced by alterations in the actions of hypothalamic appetite-regulatory factors, reductions in energy expenditure, insulin resistance in skeletal muscle, and β-cell dysfunction. Interestingly, these unfavorable effects of androgens on metabolic and body-weight regulation systems are neither observed nor evident in ovariectomized animals and post-menopausal women, indicating that the adverse effects of androgens might be dependent on the estrogen milieu. Recent findings may provide novel sex- and age-specific strategies for treating metabolic diseases.

## 1. Introduction

It is well known that reproductive function and energy balance are closely linked in most species [1,2,3]. Low energy availability, such as that associated with undernutrition and obesity, disturbs female reproductive function, and some central and peripheral factors are known to affect these changes; for example, the upregulation of hypothalamic appetite-regulatory factors, such as orexin and neuropeptide-Y (NPY), suppresses gonadotrophin-releasing hormone (GnRH) secretion. Conversely, gonadal steroid hormones (i.e., estrogens and androgens) also affect the regulation of appetite, energy metabolism, and body weight regulation in mammals and humans [1,4].

Estrogens, most of which are secreted by the ovary, generally have beneficial effects on female metabolic and body-weight regulation systems (Table 1); for example, by suppressing food intake and increasing metabolic expenditure, endogenous and exogenous estrogens prevent obesity, and several kinds of metabolic disorders in females of both reproductive and post-menopausal ages [1,5,6]. Most of these effects of estrogens are mediated by changes in hypothalamic appetite-regulatory factors. For example, in female rats, estrogens suppress hypothalamic orexigenic factors, such as NPY, agouti-related peptide, prepromelanin-concentrating hormone, and orexin [7,8], and stimulate anorexigenic factors, such as proopiomelanocortin (POMC) and corticotrophin-releasing hormone [8,9]. In addition, recent studies by our own and other groups have shown that the effects of estrogens on metabolic and body weight regulation systems are partially mediated by hypothalamic oxytocin (OT), which is one of several potent anorexigenic factors [10]. For example, when hypothalamic gene expression and circulating levels of OT are decreased in ovariectomized rats, the changes in OT levels are attenuated by estrogen supplementation [11,12]. Similarly, the administration of exogenous OT has been shown to decrease appetite, body weight, and fat mass in ovariectomized and perimenopausal female rats [13,14]. In rats, hypothalamic gene expression of OT increases during estrous when serum estrogen levels are at their highest, and in healthy women, circulating levels of OT are also increased under periovulatory hyper-estrogenic conditions [12,15].

Conversely, the effects of androgens on female metabolic and body weight regulation systems and their underlying mechanisms have not been fully clarified (Table 1), even though it has been proposed that excess levels of androgens, such as those associated with polycystic ovary syndrome (PCOS) which is one of the most common endocrine/reproductive disorders in women of reproductive age, may increase the risks of obesity and some metabolic disorders in women of reproductive age. On the other hand, the effects of androgens on post-menopausal women are highly controversial; some studies have shown that circulating levels of Dehydroepiandrosterone sulfate (DHEA-S) are not related to the risk of cardiovascular disease and mortality, whereas other studies have indicated that high testosterone levels may increase the risk of cardiovascular disease and insulin resistance [16,17]. In addition, a recent study has proposed that hyperandrogenism during reproductive ages may not necessarily increase the risk of cardiovascular disease and mortality in later life [18,19], indicating that the effects of androgen on metabolic functions might change depending on age and pre- or post-menopausal status. Interestingly, contrary to the case in females, it has been shown that androgens have favorable effects on metabolic and body-weight regulation systems in males. Indeed, a deficiency in androgens predisposes males to diabetes, and androgen replacement has been shown to reduce the risk of hyperglycemia in men [20,21,22,23,24]. Similarly, an androgen deficiency increases the risk of visceral obesity, insulin resistance, and metabolic syndrome in men [20,25,26].

In this review, we comprehensively describe the roles of androgens in metabolic functions in females of reproductive and post-menopausal age. We also discuss a recently proposed mechanism for explaining the discrepancies in the effects of androgens among different reproductive ages and sexes.

## 2. Effects of Androgens in Women of Reproductive Age

### 2.1. Effects of Endogenous Androgens on Metabolic and Body Weight Regulation Systems

It has been shown that endogenous androgens affect the regulation of metabolic and body weight regulation systems and that excess amounts of endogenous androgens may increase the risks of metabolic disturbance, such as visceral obesity, insulin resistance, diabetes, and metabolic syndrome in women of reproductive age [1]. PCOS is one of the most common endocrine/reproductive disorders in women of reproductive age, and the risks of metabolic-related disorders, such as obesity, type 2 diabetes, metabolic syndrome, and cardiovascular disease, are increased in women with PCOS compared to healthy women without PCOS [27,28,29,30]. The etiology of this syndrome remains largely unknown, and there are no corresponding studies in animals, but mounting evidence has shown that androgen excess may play a key role in the etiology and pathophysiology of PCOS [31,32,33]. In particular, hyperandrogenic status is strongly associated with the metabolic disorders typically associated with PCOS; for example, hyperandrogenic women with PCOS have higher prevalence rates of insulin resistance, abdominal obesity, and adverse metabolic profiles than women without PCOS and those with non-hyperandrogenic PCOS [27,28,34], indicating that endogenous androgen excess adversely affects the metabolic profile of women. However, it has also been reported that the adverse effects of endogenous androgen excess on metabolic and body weight regulation systems are also observed in women without PCOS. Specifically, pre-menopausal women with high free androgen indices (FAI) display higher waist circumference values, fasting glucose and insulin levels, systolic and diastolic blood pressure levels, serum triglyceride levels, and lower serum high-density lipoprotein cholesterol levels [35]. In addition, a previous study showed that women with idiopathic hirsutism displayed greater waist–hip ratios (WHR) than age- and weight-matched non-hirsute women and that their plasma testosterone levels correlated positively with WHR [36]. Further, an increase in the percentage of free testosterone and a decrease in plasma sex hormone-binding globulin capacity has been shown to be accompanied by increases in WHR, abdominal fat, and plasma insulin and glucose levels, indicating that an increase in unbound androgens may induce enlargement of abdominal adipocytes and disturb glucose–insulin homeostasis [37].

### 2.2. Effects of Blockade of Androgens on Metabolic and Body Weight Regulation Systems

Based on the findings of these studies, some studies examined the effects of antiandrogenic treatments on the metabolic status of women with PCOS. As a result, treatment with the antiandrogenic drug flutamide for 12 months reduced visceral and subcutaneous fat and improved lipid profiles and insulin sensitivity in overweight–obese women with PCOS [38]. Treatment with flutamide had additional favorable effects, such as reductions in visceral fat and low-density lipoprotein (LDL) cholesterol, when compared to obese women with PCOS fed a hypocaloric diet [39]. Similarly, administration of spironolactone, another antiandrogenic drug, also appeared to improve insulin resistance in women with PCOS [40]. These favorable effects of antiandrogenic treatments on metabolic and body weight regulation systems have been reproduced in an animal PCOS model; for example, administration of flutamide reduces body weight and adipocyte size in a letrozole-induced PCOS mice model (produced using an aromatase inhibitor), concomitantly with the attenuation of hyperandrogenemia and restoration of the estrous cycle [41]. Altogether, these findings illustrate the possibility that androgen signaling plays a key role in the metabolic phenotype of PCOS and that androgen excess might increase the risk of metabolic disorders.

### 2.3. Effects of Endogenous Androgens on Appetite

Interestingly, several studies have suggested that androgens might affect food cravings and food preferences in young, overweight, and obese women. For example, young women with high FAI exhibited increased craving for certain foods, such as high-fat foods and fast foods, independent of age, body mass index, and PCOS status [42], and women with PCOS displayed higher prevalence rates of abnormal eating behaviors, such as bulimic behavior [43]. Similarly, PCOS women with hyperandrogenism, such as high DHEA-S, testosterone, and androstenedione levels, have an increased risk of depression and food craving [44]. These findings were confirmed in a meta-analysis that included 470 women with PCOS and 390 controls; the risk of eating disorders was higher in women with PCOS compared to the controls [45]. In addition, women with bulimia nervosa tended to have higher androgen levels than normal women [46], and antiandrogenic oral contraceptive use reduced meal-related hunger and eating behavior in bulimic women [47]. Similarly, it has been reported that administration of flutamide with or without the serotonin reuptake inhibitor citalopram reduces binge eating in patients with bulimia nervosa, indicating that antiandrogen treatment may be a candidate for treating eating disorders in women [48].

### 2.4. Effects of Exogenous Androgens on Metabolic and Body Weight Regulation Systems

The hypothesis that androgen excess is related to various metabolic disorders has been supported by studies that evaluated the effects of exogenous androgens on metabolic functions. Androgen-treated female-to-male transsexual patients exhibited higher homeostatic model assessment of insulin resistance values than those who did not receive hormonal treatment [49], indicating that administration of exogenous androgens might increase insulin resistance. Conversely, another study showed that testosterone-administered female-to-male transsexual patients exhibit high triglycerides and low high-density lipoprotein levels, indicating that exogenous androgen has significant effects on lipid profiles in women [50]. In addition, the administration of methyltestosterone reduced glucose uptake during insulin infusions in non-obese pre-menopausal women, suggesting that hyperandrogenemia induces insulin resistance [51]. These phenomena can be reproduced in experimental animals by the chronic administration of androgens. For example, chronic administration of dihydrotestosterone (DHT) (4 weeks) increased visceral adiposity in female mice of reproductive age [52], and the administration of testosterone (around 2 weeks) increased visceral and subcutaneous fat weight and body weight in female rats of reproductive age [53,54]. Interestingly, these effects of androgen on fat and body weight are observed in gonadally intact and estradiol-supplemented ovariectomized rats but not in non-supplemented ovariectomized rats [53,54]. Animals that were subjected to longer-term androgen treatment (5 to 19 weeks) displayed increased food intake and body weight gain, insulin resistance, and hyperlipidemia, as well as irregular estrous cycles and morphological ovarian changes [33,55,56,57]. In addition, the chronic administration of testosterone also increased the preference of female rats of reproductive age for a high-fat diet [58], indicating that the effects of androgens on feeding behavior, e.g., hyperandrogenic young women exhibiting an increased craving for high-fat foods and fast foods, can also be reproduced in animal models. Interestingly, animals that have received longer-term androgen administration show PCOS-like phenotypes, such as increased body weight, food intake, fat mass, metabolic disturbance, and reproductive dysfunction. Recently, these animal models have been used to evaluate the etiology and pathophysiology of PCOS and to develop novel treatments for the condition.

## 3. Effects of Androgens in Women of Post-Menopausal Age

As noted above, the effects of androgens on metabolic functions in post-menopausal women are controversial. Some studies have implied that androgens have unfavorable effects on metabolic functions in post-menopausal women. For example, it has been shown that post-menopausal women aged 57–59 years with impaired glucose tolerance have higher androgen activity, i.e., lower sex hormone-binding globulin (SHBG) levels and higher testosterone-to-SHBG ratios than normal, glucose-tolerant post-menopausal women [59]. In addition, high testosterone levels in post-menopausal women aged 55–89 years have been used to predict insulin resistance and the future incidence of type 2 diabetes, i.e., bioavailable testosterone was positively related to fasting and post-challenge glucose and insulin levels [60]. Interestingly, higher serum estradiol levels are also strongly related to metabolic disorders in post-menopausal women. For example, it has been shown that bioavailable estradiol is positively related to insulin resistance in women aged 57–59 years and that post-menopausal women with a mean age of 60.3 years with relatively high plasma estradiol and testosterone levels have a highly increased risk of type 2 diabetes [61]. As opposed to endogenous androgens, it has been reported that exogenous androgens may have favorable effects on metabolic and other functions in postmenopausal women. For example, testosterone treatment has favorable effects on body composition, bone, cardiovascular function, and cognitive performance and improves sexual desire, arousal, and orgasm in post-menopausal women [62].

## 4. Mechanisms Underlying the Effects of Androgens on Metabolic and Body Weight Regulation Systems

### 4.1. Effects of Androgens on Hypothalamic Functions

It has been shown that an excess of androgens alters the activity and mRNA expression of some hypothalamic factors and that these alterations might be involved in the unfavorable effects of androgens on metabolic and body weight regulation systems. Chronic administration of DHT has been shown to reduce the hypothalamic mRNA expression of POMC, which is an anorectic factor, and to promote visceral adiposity in female mice of reproductive age [52]. It has been reported that mRNA expression of hypothalamic orexigenic factors, such as NPY and Agouti-related protein (AgRP), shows sexual dimorphism, i.e., expression levels of these factors in males were higher than those in females in avians [63]. It is possible that androgens or estrogens might be related to these sex differences. However, further studies would be needed to clarify this hypothesis. Similarly, we have shown that chronic testosterone administration reduced the hypothalamic mRNA expression of estrogen receptor-α (ER-α) and/or increased the mRNA expression of pro-inflammatory cytokines, concomitantly with the increases in food intake, body weight, and fat mass in gonadally intact female rats and/or estradiol-supplemented ovariectomized female rats [53,54]. It has been established that estrogens have favorable effects on metabolic and body weight regulation systems in females, and those favorable effects of estrogens are partly mediated by hypothalamic ER-α [1,5,6]. In addition, it has been reported that low-grade hypothalamic inflammation undermines homeostatic responses that protect against obesity [64]. Therefore, the downregulation of ER-α expression and the upregulation of pro-inflammatory cytokine expression in the hypothalamus, which were observed in gonadally intact and estradiol-supplemented rats, might underlie androgen-induced metabolic disorders. Interestingly, such alterations in hypothalamic ER-α and pro-inflammatory cytokine expression, as well as increases in fat mass, in gonadally intact or estradiol-supplemented ovariectomized rats were not observed in ovariectomized rats [53,54]. Taken together, we propose that the effects of androgens on metabolic functions might depend on the estrogen milieu; androgens have unfavorable effects on females of reproductive age, whereas their effects might be decreased or reversed in post-menopausal females.

Recently, we proposed that hypothalamic OT also be involved in the effects of androgens on metabolic and body weight regulation systems. OT is synthesized in the hypothalamus and secreted from the posterior lobe of the pituitary gland. The magnocellular OT neurons project into the posterior pituitary gland and secrete OT into the peripheral circulatory system, while the parvocellular OT neurons project and secrete OT into several regions of the central nervous system. In addition to well-established roles in the regulation of labor and lactation in female mammals [65], recent studies have shown that OT also has behavioral, psychological, and physiological functions, including metabolism, appetite, and body weight regulation [13,66,67]. In a previous study, we showed that serum OT levels were decreased in long-term DHT-administered PCOS model rats and that supplementation of OT attenuated food intake and body weight in these animals, indicating that OT might be related to the effects of androgens on appetite and body weight regulation [68]. Interestingly, the effects of estrogens on OT are completely the opposite of those of androgens. For example, we have shown that hypothalamic OT gene expression levels and serum OT levels were decreased in ovariectomized obese rats and that supplementation of estradiol in these animals restored both hypothalamic and serum OT levels [12]. Similarly, serum OT levels were increased under hyper-estrogenic conditions, i.e., during the preovulatory phase and during ovarian stimulation, in healthy women [15]. These findings indicate that, in terms of the metabolic and body weight regulation systems in females, OT might have the opposite effect of both androgens and estrogens.

### 4.2. Effects of Androgens on Energy Expenditure

It has been reported that androgen excess reduces energy expenditure, which is associated with visceral obesity. In female mice of reproductive age, chronic DHT administration prevented the leptin-induced activation of brown adipose tissue (BAT) thermogenesis, which reduced energy expenditure and increased visceral fat mass [62]. Similarly, chronic DHEA administration reduced BAT activity, body temperature, and O_2_ consumption in female rats of reproductive age, whereas the transplantation of BAT from rats that had not been administered DHEA negated these DHEA-induced changes and normalized systemic insulin sensitivity [69]. In addition, numerous studies have shown that BAT has a marked effect on the manifestation of PCOS symptoms and that increases in BAT mass or activity by transplantation and compound activation may be effective in the treatment of PCOS [70]. These results indicate that reductions in endogenous BAT activity are closely related to the adverse effects of androgens on metabolic functions.

### 4.3. Effects of Androgens on Skeletal Muscle Insulin Resistance and β-Cell Dysfunction

Previous studies have shown that an excess of androgens can induce insulin resistance in women and female experimental animals. In pre-menopausal women, high testosterone levels produced insulin resistance that was detected using hyperglycemic and euglycemic hyperinsulinemic clamp tests [51,71]. Such testosterone-induced insulin resistance is considered to occur mainly in skeletal muscle [51]. In female rats, the administration of testosterone also induced insulin resistance in muscles and caused reductions and increases in the relative numbers of type 1 and type 2 muscle fibers, respectively [72,73]. Although the detailed mechanisms by which androgens promote muscular insulin resistance remain unclear, it has been suggested that androgens act as aggravating factors rather than as initiators of insulin resistance in females. On the contrary, it has been shown that serum testosterone levels at the normal range are needed to maintain appropriate insulin sensitivity in male rats and that both castration and high serum testosterone levels are followed by marked insulin resistance [74]. These results suggest that androgens may be important regulatory factors of muscular insulin sensitivity and that their actions are maximized at physiological concentrations.

It has been suggested that an excess of androgens causes a predisposition towards pancreatic β-cell dysfunction by activating β-cell stressors. Hyperandrogenic women exhibit β-cell hyperfunction, which might induce secondary β-cell dysfunction [75]. In addition, excess testosterone in female mice induced β-cell dysfunction via an androgen receptor-dependent mechanism [76], and direct androgen exposure to isolated pancreatic islets induced an impaired response to glucose stimulation [77,78]. These effects of androgens on β-cells might be attributed to systemic oxidative stress, low-grade inflammation, and/or insulin resistance [67,78].

## 5. Effects of Prenatal Undernutrition on the Metabolic Action of Exogenous Androgens after Birth

As noted above, PCOS is a common endocrine disorder in women of reproductive age, and its prevalence rate is around 5–16%. The key phenotype is chronic anovulation and irregular menses or amenorrhea. In addition, it is well known that metabolic disturbances, such as obesity, insulin resistance, type 2 diabetes mellitus, hyperlipidemia, and metabolic syndrome, are commonly associated with PCOS [79,80,81]. Although it has been speculated that genetic and environmental factors, such as lifestyle and diet habits, may be involved in the pathogenesis of PCOS [82,83,84], the etiology of PCOS remains poorly understood. Some studies using animal models have shown that prenatal hyperandrogenic condition may be a possible etiology of PCOS, i.e., androgen exposure in utero or early neonatal period induces ovulatory dysfunction, hyperandrogenism, polycystic ovaries in female mammals and some metabolic disorders in later life [55]. However, it remains unclear whether this hypothesis can be applied to humans. It has been shown that testosterone and androstenedione levels in pregnant PCOS women were higher than those in non-PCOS patients from mid-pregnancy onwards [85]. On the contrary, another human study evaluating the relationship between early life androgen exposure and PCOS in adolescence revealed that maternal androgens did not directly program PCOS in the offspring [86].

For a long time, epidemiological, clinical, and experimental studies have shown that undernutrition in utero alters physiological functions and increases the risk of some metabolic-related diseases in adulthood. This hypothesis is called developmental origins of health of disease (DOHaD), and this may have important medical and socioeconomic implications for the prevention of diseases [87]. Recently, it has been proposed that the concept of DOHaD can be applied to the evaluation of PCOS etiology, as metabolic derangement with developmental origins is a cardinal feature of PCOS [88,89]. One study evaluated the prevalence of PCOS in women born as small for gestational age (SGA) [88]. As a result, the prevalence of PCOS was twice as high in women born as SGA than in women born as appropriate for gestational age (AGA). Similarly, another study evaluated the relationship between birth characteristics and PCOS symptoms [90]. As a result, this study revealed that each 100 g increase in birth weight increased the risk of hyperandrogenism by 5%, supporting the hypothesis that prenatal undernutrition may be linked to the etiology and/or pathophysiology of PCOS. On the contrary, some studies could not find the relationship between birth weight and the phenotypes of PCOS. Namely, one study evaluated the association between body weight at birth and the symptoms of PCOS. As a result, both birth weight and SGA were not related to the POCS symptoms in adulthood [88]. Another study examined the relationship between birthweight and gestational age and phenotypes of PCOS and showed that birthweight had no substantive association with metabolic and reproductive phenotypes of PCOS [90]. Most of these studies evaluated the association between birth weight and PCOS phenotypes, and no study had examined the relationship between prenatal undernutrition, which is the key factor of DOHaD, and the onset and phenotype of PCOS after birth. Because the effects of bias and confounding variables cannot be completely eliminated from epidemiological and retrospective clinical studies, and reproductive and metabolic organs cannot be explored in human studies, experimental studies using animal models are useful for the evaluation of the etiology and pathophysiological mechanisms of PCOS.

Regarding the DOHaD, metabolic and reproductive changes caused by prenatal undernutrition can be reproduced in some experimental models. Namely, mice and rats exposed to prenatal undernutrition exhibit obesity and insulin resistance in adulthood. In addition, female rats exposed to prenatal undernutrition show delayed puberty by the decrease of hypothalamic kisspeptin actions [3]. Similarly, female rats exposed to prenatal undernutrition show ovaries with large cystic structures, reduced corpora lutea, and premature reproductive senescence [91]. As noted above, animals that have received longer-term androgen administration show PCOS-like phenotypes, and these animal models have been used to evaluate the etiology and pathophysiology of PCOS. Among these PCOS models, chronically DHT-administered mice and rats are commonly used as they reproduce the typical features of PCOS, such as obesity, adiposity, anovulation, and atypical follicles [32,55]. Based on the above background, we proposed the hypothesis that DOHaD might be involved in the etiology or pathophysiology of PCOS and evaluated the effects of prenatal undernutrition on the phenotypes of the PCOS model induced by chronic DHT administration [57]. As a result, prenatal undernutrition exacerbated the DHT-induced metabolic change, i.e., it increased body weight gain, food intake, fat weight, and adipocyte size in PCOS model rats. Prenatal undernutrition also exacerbated the upregulation of inflammatory cytokines in adipose tissue and increased the gene expressions of hypothalamic orexigenic factor. On the contrary, prenatal undernutrition attenuated the DHT-induced reproductive change, i.e., it attenuated the irregular cyclicity, polycystic ovaries, and disturbed gene expression of ovarian steroidogenic enzymes. In addition, prenatal undernutrition did not affect metabolic and reproductive functions in non-DHT-administered control rats. These results indicate that prenatal undernutrition alters the phenotype of the DHT-induced PCOS model, i.e., it exacerbates the metabolic phenotypes and attenuates the reproductive phenotypes, whereas prenatal undernutrition itself cannot produce the PCOS phenotypes without the postnatal hyperandrogenic condition. From an evolutionary perspective, it has been speculated that PCOS might provide a survival advantage under prehistoric conditions of undernutrition [92,93]. Insulin resistance can protect from protein loss and increase the survival rate under prolonged starvation, and reduced fecundity due to ovulatory disorders might provide a survival advantage. Because genetic studies revealed that PCOS may have existed 50,000 years ago, the genetic factors related to PCOS might be beneficial for survival under past harsh environments [93]. It is assumed that these PCOS-related mechanisms cannot be adapted to the rapid improvement of nourished conditions and that they became pathogenic factors of metabolic and reproductive diseases during the past few decades. Our data suggests that not only long-term evolutional changes but also changes with DOHaD may modify the PCOS phenotype; namely, prenatal undernutrition exacerbates the metabolic phenotypes of PCOS to survive the expected starved condition after birth. However, more precise experimental studies and epidemiological and clinical studies with large sample sizes are needed to clarify this hypothesis.

## 6. Conclusions

This review highlights the effects of androgens on metabolic and body weight regulation systems and their underlying mechanisms. Excess levels of endogenous and exogenous androgens in reproductive age can induce several unfavorable effects on metabolic functions and increase the risks of metabolic-related diseases. Basic studies using laboratory animals indicate that some central and peripheral factors and changes in physiological functions might be related to these adverse effects of androgens (Figure 1). Conversely, the effects of androgens on metabolic and body weight regulation systems in females of post-menopausal age are highly controversial.

## Figures and Tables

**Figure 1 ijms-24-16567-f001:**
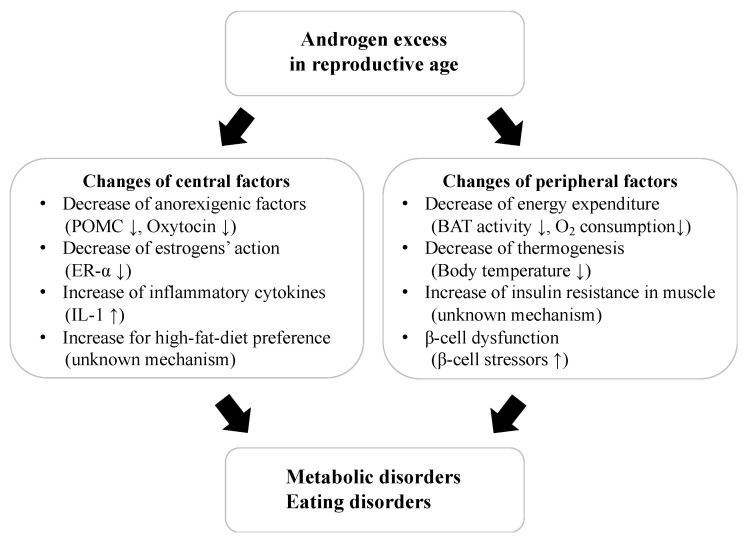
The effects of androgen excess on metabolic and body weight regulations in reproductive age. POMC; proopiomelanocortin, ER; estrogen receptor, IL; interleukin, BAT; brown adipose tissue.

**Table 1 ijms-24-16567-t001:** The metabolic effects of androgens and estrogens in females and males.

		Androgens	Estrogens
Female	Reproductive age	Adverse effects	Favorable effects
	Post-menopausal age	Controversial	Favorable effects
Male		Favorable effects	

## Data Availability

Data are contained within the article.

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
