# Peer review of "Age-Dependent Changes in the Effects of Androgens on Female Metabolic and Body Weight Regulation Systems in Humans and Laboratory Animals"

_ijms, 2023, doi:10.3390/ijms242316567_

Round 1

Reviewer 1 Report

Comments and Suggestions for Authors

The review paper by Takeshi Iwasa et al. provides a comprehensive analysis of the effects of androgens on metabolic functions in females of reproductive and postmenopausal age, with a notable emphasis on PCOS-like phenotypes.  This well-written manuscript delves into the intricate details of these effects, offering valuable insights for researchers and healthcare professionals.

Given the primary focus on females, it is suggested that the title of the paper can omit reference to males, making it more concise.

Minor points

Line 207 »…rats and/or estradiol-supplemented supplemented ovariectomized female rats…« please delete repeated word.

Line 246 »O2« please change to »O2«

Line 312 »...showed that birthweight in had no substantive…« »in« seems to be surplus.

Line 353 »…PCOS-related mechanisms cannot be adapt to the rapid improvement …« – please change to »…PCOS-related mechanisms cannot be adapted to the rapid improvement …«.

Line 331 »atypical follicle« – suggestions »atypical follicles«.

Figure 1 B-»cell dysfunction« – suggestions »β-cells dysfunction«.

Author Response

General comment

The review paper by Takeshi Iwasa et al. provides a comprehensive analysis of the effects of androgens on metabolic functions in females of reproductive and postmenopausal age, with a notable emphasis on PCOS-like phenotypes.  This well-written manuscript delves into the intricate details of these effects, offering valuable insights for researchers and healthcare professionals.

Given the primary focus on females, it is suggested that the title of the paper can omit reference to males, making it more concise. 

Response

Thank you for your valuable comments for our manuscript. In accordance with your comment, we have revised manuscript. We believe that our manuscript may be improved compared with previous version and would be accepted. The term of ‘male’ is omitted in Title. In addition, in accordance with comment from another reviewer, title is changed as ‘Age dependent changes in the effects of androgens on female metabolic and body weight regulation systems in human and laboratory animals’ in revised manuscript.

Minor points

Comment

  1. Line 207 »…rats and/or estradiol-supplemented supplemented ovariectomized female rats…« please delete repeated word.
  2. Line 246 »O2« please change to »O2«
  3. Line 312 »...showed that birthweight in had no substantive…« »in« seems to be surplus.
  4. Line 353 »…PCOS-related mechanisms cannot be adapt to the rapid improvement …« – please change to »…PCOS-related mechanisms cannot be adapted to the rapid improvement …«.
  5. Line 331 »atypical follicle« – suggestions »atypical follicles«.
  6. Figure 1 B-»cell dysfunction« – suggestions »β-cells dysfunction«.

Response

Thank you for your precise advised. We have corrected all the points you pointed out in our revised manuscript.

Reviewer 2 Report

Comments and Suggestions for Authors

In the present manuscript, Takeshi Iwasa et al. reviewed the differential effects of androgens on metabolism and body weight regulation systems among females of reproductive age, women of postmenopausal age, and males. The paper is interesting and correctly written, but there are some important issues that should be addressed.

I address some of these below:

1.           There are 21 authors of this review paper. In my opinion, this is too many considering the quality and complexity of this paper.

2.           I think that the authors trying to analyze women of reproductive age, women of postmenopausal age, and males include too many groups, which may complicate the interpretation of the results. In my opinion, the group of men is unnecessary and disturbs the clarity of the results.

3.           If the authors decide to include men in the analysis, it would be useful to clearly define the common reference points and the specific aspects of androgen influence that they wish to compare between the sexes. This may require more complex methods of analysis and the inclusion of existing studies.

4.           The abstract does not mention males because the comparison is between humans (women) and animals (rodents).

5.           The title of the study is very general. In the context of all the information presented, the title could be more precise to better reflect the specificity of the research and the diversity of the study groups. If the study was actually a comparison of different human groups (women at different life stages, men) and laboratory animals, the title might better reflect this.

6.           The researchers point to a very broad approach in the study that includes multiple groups of humans and animals, transgender individuals, etc. that cannot be accurately compared, leading to cognitive chaos.

-             - Studying humans, animals, and different groups of people (e.g., transgender individuals, males, females in different age groups) may complicate interpretation of results.

-             - Animals such as mice and rats have different hormonal systems and metabolism than humans. When the results of animal studies are combined with human data in an analysis, inaccurate conclusions may result.

-             - Transgender individuals face unique social and medical challenges that may impact their metabolic health. When their data are mixed with those of other groups, certain unique aspects of this group may be obscured.

7.           If the reader wishes to understand the general effects of androgens on metabolism and body mass, such a comprehensive analysis is warranted. However, if one wishes to understand mechanisms in specific groups, it would be more beneficial to focus on a smaller number of groups. A broad approach can provide a wealth of data, but there is a risk that specific and clear conclusions cannot be drawn because of the excessive number of variables and study groups. This can be confusing for the reader. Interpretation of the results is complex.

8.           There is no mention of males in the abstract, but there is a reference to animals that is also not mentioned in the title. At this point, it would be appropriate to include a sentence about males (that androgens have a different effect on these systems in males) or to include the human-animal context in the title.

9.           The terms humans, women and men, males, females are apparently not used consistently.

10.         The authors should explain what PCOS is and point out that there are no corresponding studies in animals

11.         The titles of subsequent chapters and subsections also indicate a lack of consistency and coherence in their choice of names. The researchers seem to jump from one topic to another.

12.         In reviewing the bibliography, it's also noticeable that there are some rather old studies. The most recent are from 2021/22, and there aren't many of those. There are many studies from the 90s and even the 80s.

13.         In this review, the influence of androgens on the regulatory systems of metabolism and body mass and the underlying mechanisms were highlighted. There is little explanation of how these mechanisms work, both in the text and graphically.

14.         The authors' conclusions aren't precise and accurately comparative-in this case, comparing humans and animals with postmenopausal women is highly controversial.

15.         The title needs improvement - it doesn't fully reflect what is happening in the study.

16.         Some of the text didn't specify exactly which groups of people were studied (e.g., whether they were healthy people or people with certain diseases)

17.         The text also referred to laboratory animals. It would be useful to mention which animals were studied and how the results can be applied to humans.

18.         It's important that the authors closely control for factors that may influence the results and clearly communicate their conclusions by highlighting differences and similarities between the groups-but only those that are meaningful for comparison.

19.         The mention of "men" in the title of the study isn't appropriate because men aren't significantly addressed in the content. The focus is primarily on women and rodents.

20.         Comparing women of reproductive age to postmenopausal women and males may be difficult, particularly with respect to the influence of hormones on metabolism and body mass, as each of these groups has a different hormone profile that may influence different physiological aspects. It's suggested to analyze only the women and to mention the men only briefly.

Author Response

Response to Reviewer 2

General comment

In the present manuscript, Takeshi Iwasa et al. reviewed the differential effects of androgens on metabolism and body weight regulation systems among females of reproductive age, women of postmenopausal age, and males. The paper is interesting and correctly written, but there are some important issues that should be addressed.

 Response

Thank you for your valuable comments for our manuscript. In accordance with your comment, we have revised manuscript. We believe that our manuscript may be improved compared with previous version and would be accepted.

Comment

There are 21 authors of this review paper. In my opinion, this is too many considering the quality and complexity of this paper.

Response

All authors made some contribution to this paper and cited research. Thus, I want to list them as far as possible.

Comment

I think that the authors trying to analyze women of reproductive age, women of postmenopausal age, and males include too many groups, which may complicate the interpretation of the results. In my opinion, the group of men is unnecessary and disturbs the clarity of the results.

If the authors decide to include men in the analysis, it would be useful to clearly define the common reference points and the specific aspects of androgen influence that they wish to compare between the sexes. This may require more complex methods of analysis and the inclusion of existing studies.

The abstract does not mention males because the comparison is between humans (women) and animals (rodents).

Response

Thank you for your suggestion. In accordance with your recommendation, most explanation about male was excluded in revised manuscript.

Comment

The title of the study is very general. In the context of all the information presented, the title could be more precise to better reflect the specificity of the research and the diversity of the study groups. If the study was actually a comparison of different human groups (women at different life stages, men) and laboratory animals, the title might better reflect this.

Response

In accordance with your comment, title is changed as ‘Age dependent changes in the effects of androgens on female metabolic and body weight regulation systems in human and laboratory animals’ in revised manuscript.

Comment

The researchers point to a very broad approach in the study that includes multiple groups of humans and animals, transgender individuals, etc. that cannot be accurately compared, leading to cognitive chaos. -             - Studying humans, animals, and different groups of people (e.g., transgender individuals, males, females in different age groups) may complicate interpretation of results. -             - Animals such as mice and rats have different hormonal systems and metabolism than humans. When the results of animal studies are combined with human data in an analysis, inaccurate conclusions may result. -             - Transgender individuals face unique social and medical challenges that may impact their metabolic health. When their data are mixed with those of other groups, certain unique aspects of this group may be obscured. If the reader wishes to understand the general effects of androgens on metabolism and body mass, such a comprehensive analysis is warranted. However, if one wishes to understand mechanisms in specific groups, it would be more beneficial to focus on a smaller number of groups. A broad approach can provide a wealth of data, but there is a risk that specific and clear conclusions cannot be drawn because of the excessive number of variables and study groups. This can be confusing for the reader. Interpretation of the results is complex.

It's important that the authors closely control for factors that may influence the results and clearly communicate their conclusions by highlighting differences and similarities between the groups-but only those that are meaningful for comparison.

Response

As you pointed out, this review discusses the broad information about the androgens’ effects from many aspects, such as human of reproductive age and postmenopausal age, transgender, and laboratory animal. Although, it is good idea that these data are discussed and compared separately, they may be slightly difficult as data size of each group is not so large. Thus, we want to select the broad approach in this review to show the wealth data.

Comments

There is no mention of males in the abstract, but there is a reference to animals that is also not mentioned in the title. At this point, it would be appropriate to include a sentence about males (that androgens have a different effect on these systems in males) or to include the human-animal context in the title.

Response

In accordance with your comment, as noted above, title is changed as ‘Age dependent changes in the effects of androgens on female metabolic and body weight regulation systems in human and laboratory animals’ in revised manuscript.

Comments

The terms humans, women and men, males, females are apparently not used consistently.

Response

In this review, women and men are used for human of each sex. The term of human is used inclusively for men and women. Males and females are used for laboratory animals of each sex. As you pointed out, mixed use of these terms is slightly confusion, and some terms have been replaced with another words in revised manuscript.

Comment

The authors should explain what PCOS is and point out that there are no corresponding studies in animals.

Response

Explanation about the PCOS is stated as following, ‘one of the most common endocrine/reproductive disorders in women of reproductive age, and the risks of metabolic-related disorders, such as obesity, type 2 diabetes, metabolic syndrome, and cardiovascular disease, are increased in women with PCOS’. In addition, explanation about that there are no corresponding studies in animals is stated.

Comment

The titles of subsequent chapters and subsections also indicate a lack of consistency and coherence in their choice of names. The researchers seem to jump from one topic to another.

Response

As you pointed out, the titles were lack of consistency. Thus, we changed some titles as following, ‘Effects of blockade of androgens on metabolic and body weight regulation systems’, ‘Effects of endogenous androgens on appetite’, ‘Effects of androgens on hypothalamic functions’, ‘Effects of androgens on energy expenditure’, ‘Effects of androgens on skeletal muscle insulin resistance and β-cell dysfunction’.

Comment

In reviewing the bibliography, it's also noticeable that there are some rather old studies. The most recent are from 2021/22, and there aren't many of those. There are many studies from the 90s and even the 80s.

Response

As you pointed out, references of this review are rather old. Although we tried to find recent studies, there are only a few papers discussing the effects of androgens on female metabolic function (they have been already cited). We think that this area is important and should be evaluated more vigorously, and hope that present review might encourage other researchers.

Comment

In this review, the influence of androgens on the regulatory systems of metabolism and body mass and the underlying mechanisms were highlighted. There is little explanation of how these mechanisms work, both in the text and graphically.

Response

The underlying mechanism of androgens effects on female metabolism have not been well established. As far as our knowledge, information shown in fourth section can cover the most past studies. We revised Figure as showing specific factors involved in these mechanisms.

Comment

The authors' conclusions aren't precise and accurately comparative-in this case, comparing humans and animals with postmenopausal women is highly controversial.

Response

As you pointed out, androgens effects of postmenopausal women is highly controversial, thus, we modified our conclusions and deleted the following sentence ‘It is possible that differences in the estrogen milieu might be related to these discrepancies in women of pre- and postmenopausal ages’.

Comment

Some of the text didn't specify exactly which groups of people were studied (e.g., whether they were healthy people or people with certain diseases). The text also referred to laboratory animals. It would be useful to mention which animals were studied and how the results can be applied to humans.

Response

In accordance with your comments, we specify which groups of people were studied in cited reference. We also specified the rodents (rat or mice) used for each study throughout the revised manuscript.

Comment

The mention of "men" in the title of the study isn't appropriate because men aren't significantly addressed in the content. The focus is primarily on women and rodents.

Response

As noted above, we changed the title of this review as ‘Age dependent changes in the effects of androgens on female metabolic and body weight regulation systems in human and laboratory animals’.

Comment

Comparing women of reproductive age to postmenopausal women and males may be difficult, particularly with respect to the influence of hormones on metabolism and body mass, as each of these groups has a different hormone profile that may influence different physiological aspects. It's suggested to analyze only the women and to mention the men only briefly.

Response

Thank you for your suggestion. In accordance with your recommendation, most explanation about male was excluded in revised manuscript.

Reviewer 3 Report

Comments and Suggestions for Authors

The review article is devoted to the analysis of the problem of the influence of androgens on the regulation of metabolism and body weight in women before and after menopause. The problem is urgent. The review article is interesting and important.

My comments:

1. Men should be removed from the title of the article, since the article discusses the problem of the influence of androgens on the female body.

2. The main conclusions of the article, indicated in the abstract and in the conclusion, are summarized in the differences in the influence of androgens on the regulation of metabolism and body weight in women before and after menopause. These significant differences should be included in the final figure or table.

3. The analyzed articles from 2023 must be additionally included in the list of references. So far there are 3 of them, but more are needed, since the article is planned for publication not at the beginning, but at the end of 2023.

Author Response

Response to Reviewer 3

General comment

The review article is devoted to the analysis of the problem of the influence of androgens on the regulation of metabolism and body weight in women before and after menopause. The problem is urgent. The review article is interesting and important.

Response

Thank you for your valuable comments for our manuscript. In accordance with your comment, we have revised manuscript. We believe that our manuscript may be improved compared with previous version and would be accepted.

My comments:

Comment

Men should be removed from the title of the article, since the article discusses the problem of the influence of androgens on the female body.

Response

In accordance with your comment, as noted above, title is changed as ‘Age dependent changes in the effects of androgens on female metabolic and body weight regulation systems in human and laboratory animals’ in revised manuscript.

Comment

The main conclusions of the article, indicated in the abstract and in the conclusion, are summarized in the differences in the influence of androgens on the regulation of metabolism and body weight in women before and after menopause. These significant differences should be included in the final figure or table.

Response

As shown in Figure 1, effects of androgens on metabolic functions and their underlying mechanisms in female of reproductive age has been well established. On the other hand, their effects in postmenopausal age are highly controversial (this is also pointed out by another reviewers). Thus, it may be difficult to summarize in Figure.

Comment

The analyzed articles from 2023 must be additionally included in the list of references. So far there are 3 of them, but more are needed, since the article is planned for publication not at the beginning, but at the end of 2023.

Response

As you pointed out, references of this review are rather old. Although we tried to find recent studies, there are only a few papers discussing the effects of androgens on female metabolic function (they have been already cited). We think that this area is important and should be evaluated more vigorously, and hope that present review might encourage other researchers.

Reviewer 4 Report

Comments and Suggestions for Authors

The review by Iwasa et al succinctly explains the role of androgens on metabolic parameters in both males and females. The review is well-written and reads in a very structured and organized manner. For the most part, the review is exhaustive with only a few aspects needing more detail. I only have a few comments for them-

1. Since the focus is on body weight and energy homeostasis, it could be useful to write more on their roles in altering the activity of various neuronal populations in the arcuate nucleus. You mentioned a sentence or two on POMC neurons, but nothing much on AGRP. This is especially important because both these neuronal populations exhibit sexual dimorphism. This has been stressed in several recent publications and reviews. 

2. How about the role of androgens on regulating the HPT axis? Especially from a viewpoint of energy expenditure?

3. It seems that androgens can affect glucose and energy homeostasis via a multitude of pathways by altering the functions of various metabolic tissues and endocrine glands. A schematic describing all these effects, via its actions on various organs, could help in making this information more appealing to the readers. Additionally, a table that discerns the various effects in males vs females could also be included. 

Author Response

Response to Reviewer 4

General comment

The review by Iwasa et al succinctly explains the role of androgens on metabolic parameters in both males and females. The review is well-written and reads in a very structured and organized manner. For the most part, the review is exhaustive with only a few aspects needing more detail. I only have a few comments for them.

Response

Thank you for your valuable comments for our manuscript. In accordance with your comment, we have revised manuscript. We believe that our manuscript may be improved compared with previous version and would be accepted.

Comment

Since the focus is on body weight and energy homeostasis, it could be useful to write more on their roles in altering the activity of various neuronal populations in the arcuate nucleus. You mentioned a sentence or two on POMC neurons, but nothing much on AGRP. This is especially important because both these neuronal populations exhibit sexual dimorphism. This has been stressed in several recent publications and reviews. 

Response

In accordance with our advice, we added the explanation about the sexual dimorphism of NPY and AgRP by cited one new paper as following, ‘It has been reported that mRNA expression of hypothalamic orexigenic factors, such as NPY and Agouti-related protein (AgRP), shows sexual dimorphism, i.e. expression levels of these factors in males were higher than those in females in avian [60]. It is possible that androgens or estrogens might be related in these sex difference, however, further studies would be needed to clarify this hypothesis’.

Comment

How about the role of androgens on regulating the HPT axis? Especially from a viewpoint of energy expenditure?

Response

As far as our knowledge, there are no studies that evaluating the role of androgens on regulation of HPT axis.

Comment

It seems that androgens can affect glucose and energy homeostasis via a multitude of pathways by altering the functions of various metabolic tissues and endocrine glands. A schematic describing all these effects, via its actions on various organs, could help in making this information more appealing to the readers. Additionally, a table that discerns the various effects in males vs females could also be included. 

Response

In accordance with your comment, we revised Figure as showing specific factors involved in these mechanisms. However, description about males have been excluded by another reviewer’s comment.

Round 2

Reviewer 2 Report

Comments and Suggestions for Authors

I believe that the work still brings nothing new, but after the changes made by the authors, it can be published in its present form.

Author Response

General comment

I believe that the work still brings nothing new, but after the changes made by the authors, it can be published in its present form.

Response

Thank you for your comments for our manuscript. In revised manuscript, we added some newly published papers. We believe that this revised manuscript show some new information.

Reviewer 3 Report

Comments and Suggestions for Authors

Authors should add 2023 literature sources and discuss them

Author Response

General comment

Authors should add 2023 literature sources and discuss them

Response

Thank you for your valuable comments for our manuscript. In accordance with your comment, we have added some 2023 literature sources.